# Increasing Heavy Metal Tolerance by the Exogenous Application of Organic Acids

**DOI:** 10.3390/ijms23105438

**Published:** 2022-05-13

**Authors:** Andrea Vega, Ninoska Delgado, Michael Handford

**Affiliations:** Centro de Biología Molecular Vegetal, Departamento de Biología, Facultad de Ciencias, Universidad de Chile, Las Palmeras 3425, Ñuñoa, Santiago 7800024, Chile; andreavega.du@gmail.com (A.V.); ninoskadelgado@gmail.com (N.D.)

**Keywords:** carboxylic acid, chelator, heavy metal, phenolic acid

## Abstract

Several metals belong to a group of non-biodegradable inorganic constituents that, at low concentrations, play fundamental roles as essential micronutrients for the growth and development of plants. However, in high concentrations they can have toxic and/or mutagenic effects, which can be counteracted by natural chemical compounds called chelators. Chelators have a diversity of chemical structures; many are organic acids, including carboxylic acids and cyclic phenolic acids. The exogenous application of such compounds is a non-genetic approach, which is proving to be a successful strategy to reduce damage caused by heavy metal toxicity. In this review, we will present the latest literature on the exogenous addition of both carboxylic acids, including the Kreb’s Cycle intermediates citric and malic acid, as well as oxalic acid, lipoic acid, and phenolic acids (gallic and caffeic acid). The use of two non-traditional organic acids, the phytohormones jasmonic and salicylic acids, is also discussed. We place particular emphasis on physiological and molecular responses, and their impact in increasing heavy metal tolerance, especially in crop species.

## 1. Introduction

Heavy metals (HMs) are a group of metals and metalloids that exist naturally on the Earth’s surface. A common definition of an HM is one that has a density greater than 5 g/cm^3^ or 20 g/mol [1]; as such, 53 elements are found within this category [2]. These elements are found in rock formations or within the Earth’s surface itself and have been introduced naturally into superficial layers by geological activity such as volcanoes and sea movements over billions of years. Importantly, some of these elements fulfill critical biological functions, such as copper (Cu), chromium (Cr), iron (Fe), manganese (Mn), zinc (Zn), molybdenum (Mo), and nickel (Ni). At very low concentrations, they are thus considered essential for the correct functioning of enzymes as cofactors, receptor sites, hormonal functions, and protein transport [3]. Another group of HMs do not have a biological role, and are thus deemed non-essential, such as arsenic (As), cadmium (Cd), cobalt (Co), lead (Pb), mercury (Hg), and vanadium (Vn) [4]. All HMs constitute a potential (and sometimes hidden) threat to organisms within the ecosystem, since they can cause damage even in low concentrations, and as they are not biodegradable, HMs bioaccumulate in the food chain. The potential harm caused by HMs has increased markedly since the Industrial Revolution, as anthropogenic activity in the environment, mainly due to combustion, iron, and steel industries, exploitation of mineral resources, and the use and disposal of agrochemicals, paints, and coatings is associated with the incorporation of ever-higher levels of this group of metals [5].

Plants, being sessile organisms, are not exempt from suffering the negative effects of HM contamination. In general, it has been shown that HMs can interrupt the catalytic activity of enzymes, in addition to generating an imbalance of reactive oxygen species (ROS), thus triggering oxidative stress in plants, lipid peroxidation, and damage to photosynthetic pigments and proteins [6]. For example, it has been observed that a higher concentration of Pb in the soil reduces productivity, causing plants to have a darker green color, lower seed germination rate, accelerated fall of senescent leaves, reduced foliage, and brown roots [7]. In the case of Mn, excess concentrations lead to the appearance of chlorosis in young leaves, and necrotic spots and wrinkled mature leaves, ultimately inhibiting plant growth [8], whilst soils contaminated predominantly with Cd and Zn affect chlorophyll fluorescence parameters, growth, and development of cabbage (*Brassica oleracea*) seedlings, although a compensatory redirection of photosynthates towards mechanisms alleviating the toxic effects of the HMs may be responsible for the absence of reductions in plant biomass [9]. In general, it can be summarized that HMs act as pollutants that reduce the germination of seeds, the rate of photosynthesis, respiration, transpiration, and growth, and therefore, the quality and productivity of crops [10].

However, plants have developed internal and external mechanisms to remedy HM contamination. These organisms can concentrate pollutants in their airborne tissues and exclude HMs by biodegradation, or biotransformation into inert forms in their tissues. In addition, roots can capture these elements through the action of specific compounds with chelating activity, and consequently HMs can be neutralized, translocated, and stored [10]. Tolerance to these contaminants is given by the chelating activity of a set of Low Molecular Weight Chemical Molecules (LMWCMs), in a process in which ions/molecules of a ligand bind to atoms/ions of the central metal through a coordination bond in a cyclic or ring structure [11]. Some of these LMWCMs are amino acids, such as proline and histidine, or proteins like Phytochelatins (PCs) and Metallothioneins (MTs). However, there is a substantial group of Natural Low Molecular Weight Organic Acids (NLMWOA) such as citric acid, oxalic acid, malic acid, salicylic acid, phytic acid, lipoic acid, and phenolic acids, including the flavonoids catechin, rutin, gallic acid, and caffeic acid that also play a critical natural role in the response to HM contamination. In some cases, root exudates contain NLMWOA which lower the pH of the rhizosphere, modifying the bioavailability of nutrients, and may actually increase the solubility of essential HMs such as Zn and Fe. On the other hand, organic acids (OAs) have the ability to chelate HM cations through their carboxyl groups (-COOH) forming a complex, meaning that the effective external concentration of these toxic elements is lowered [12]. The release of OAs into the external environment can also be induced in response to HMs. On entry of the HM, the cation may activate (directly or indirectly) OA efflux channels or interact with a receptor protein, inducing the genes involved in OA biosynthesis. In some instances, OA-HM complexes formed on the root surface can enter the interior of the cell by passive diffusion or via active transport involving specific ligand/transporter channels such as NRAMP and ABC, and may be subsequently sequestered in intracellular compartments, including the vacuole [12].

Therefore, the aim of this review is to present a recapitulation of the latest advances concerning the exogenous application of selected naturally occurring organic acids (OAs) to crops, the chemical structures of which are shown in Table 1. The OAs selected are representative of those with a variety of structures (linear (citric, malic, oxalic acid); phenolic/cyclic (jasmonic, salicylic, gallic, caffeic acid); sulfur-containing (lipoic)), and include those with roles in other aspects of plant development (salicylic, jasmonic acid). In each case, their effects at morphological, physiological, and molecular levels, and how different strategies for their exogenous supplementation have been used to mitigate the effects of excess HM levels, are discussed, using recent examples from crop species of economic importance in different parts of the world.

## 2. Application of Organic Acids to Alleviate HM Stress

### 2.1. Carboxylic Acids

#### 2.1.1. Citric Acid

Citric acid (CA) is a key compound of primary metabolism, synthesized by the condensation of acetyl-CoA (2 carbons) and oxaloacetate (4 carbons) to form citrate with the intervention of the enzyme citrate synthase (CS), thus initiating the Kreb’s Cycle [13]. This molecule has an antioxidant function, but it has also been shown to have chelating activity for HMs such as Cu, Cd, Cr, Pb, Ni, Al, Hg, and As [7,14,15,16,17,18,19,20]. In general, CA and other OAs have one or several carboxyl groups which act as a ligand for HMs, thus chelating and reducing the toxicity of the HM. Once the CA-chelated complex is formed, it can be transported via the xylem to aerial parts of the plant. On translocation into the cytosol, if the concentrations of HMs reach toxic levels, they can be stored inside the vacuole and maintained in a non-toxic state by the influx of additional OAs, such as CA [21].

The exogenous application of CA in different crops improves the growth rate, a product of higher chlorophyll content and photosynthetic activity, and increases efficiency in the use of water, a greater activity of antioxidant enzymes and enhanced reduction of ROS (Table 2) [22]. In one study, tomato seeds (*Solanum lycopersicum*) were grown in soils provided with 10 μM Pb and 10 μM As. The exogenous supply of CA (250 μM) increased morphological parameters and α and β chlorophyll content, the performance of antioxidant molecules such as anthocyanins and α-tocopherol, and the efficiency of enzymes with antioxidant capacity such as catalase (CAT), ascorbate peroxidase (APX), and glutathione reductase (GR). Simultaneously, other parameters decreased, such as DNA damage and lipid peroxidation (amount of malonaldehyde, MDA, an indicator of lipid peroxidation) [18]. In oilseed rape (*Brassica napus*) grown under greenhouse conditions with Cu (50–100 μM), Cd (0.6 mM), Pb and As (10 μM), or Cr (100–500 μM), where a dose of 2.5 and 5.0 mM of CA was applied in a nutrient medium, it was observed that HM toxicity was mitigated [14,15,16]. In different crop species, such as mustard (*Brassica juncea*), nightshade (*Solanum nigram*), maize (*Zea mays*), sunflower (*Helianthus annuus*), rice (*Oryza sativa*), and alfalfa (*Medicago sativa*), the efficiency of CA (2.5 mM, 5 mM, 50 μM, 100 μM) as a chelator of Cd (0.6 mM, 300 mg kg^−1^, 50 mg, 25.0 μM), Pb, and Hg (1, 2.5, 5 mM) has also been demonstrated [19,23,24,25,26,27,28]. In calendula (*Calendula officinalis*) grown in calcareous soils enriched with Cd (50–100 mg kg^−1^), plants treated with exogenous CA in doses of 0.05 and 0.1 mM did not show any physiological symptoms related to Cd toxicity. In the same study, CA was shown to be a more efficient agent for phytoremediation compared to other chelators such as tartaric acid and ethylenediaminetetraacetic acid (EDTA) [29]. CA-mediated tolerance to HM toxicity has also been tested in forestry settings. For example, the supply of 20 mM CA in soils that were exposed to Cd (3–6 mg kg^−1^) improved parameters of basket willow (*Salix viminalis*), such as bioavailability, mobility, and distribution of Cd in the plant, as well as the amount of chlorophyll, the photosynthetic rate, and gas exchange [30]. Furthermore, in soils containing Pb (100 mg kg^−1^), foliar application and irrigation with exogenous CA (5–10 mM) of larch (*Larix olgensis*) decreased the toxic effects of Pb, raised the activities and efficiencies of peroxidase (POD) and superoxide dismutase (SOD), the amount of chlorophyll, proline, and carotenoids, and decreased lipid damage [31].

At the molecular level, effects of CA applications on gene expression can be divided into three large groups of genes, leading in unison to enhanced HM tolerance. The first group are the genes required for CA biosynthesis itself, the second set are specific transporters of HMs or metalloids, whilst the third and largest group are those responsible for responses to HM stress and which may serve as stress markers. The work of Kaur et al. [23] analyzed members of this latter group, discovering that the expression of three genes in mustard seeds—chlorophylase (*CHLASE*), phytoene synthase (*PSY*), and chalcone synthase (*CHS*) genes—all of which are linked to photosynthetic activity and are indicative of stress by HM, changed markedly when exogenous CA (100 nM) was applied directly to soil contaminated with Cd (0.6 mM CdCl_2_). Other genes involved in HM or metalloid transport are those of the NRAMP family (Natural Resistance Associated Macrophage Protein Family), whose corresponding proteins are found in the plasma membrane and have the ability to mediate the uptake of Mn and Cd. Expression of most *OsNramp* genes is increasingly inhibited by greater Cd stress (as CdCl_2_) in rice plants [32]. Foliar application of CA (5.0 mM) onto Cd-treated plants significantly increased the expression of this gene family, especially that of *OsNramp2* and *5* by up to 125%, and reduced or even eliminated the inhibition of Cd in their expression [32]. The expression of other genes, including transporters, has also been studied in the shrub *Salix variegata* exposed to Cd (50 µM in hydroponic medium). Specifically, the expression of *HMA1* (P-type metal ATPase protein 1), *PCS1* (PC synthase 1), *HMA3* (P-type metal ATPase protein 3), *Nramp5*, *MTP1* (metal tolerance protein 1), *MTP4* (metal tolerance protein 4), and *MT1A* and *MT2B* (MT 1A and 2B) in leaves was reduced by Cd-stress, a feature that was reversed (by 1.7–7.0-fold) on addition of CA (100 μM) [33].

CA has a long-known chelation effect on other metals which are not considered to be HM, such as aluminum (Al). Indeed, genetic approaches (rather than exogenous applications) have been employed with significant success to combat the effects of Al-toxicity on diverse species. For instance, the overexpression of CS from rice or yuzu (*Citrus junos*) in tobacco (*Nicotiana benthamiana*) increased internal citrate levels and enhanced tolerance to contamination by Al (40 mg/L, in nutrient solution [63]; 200 μM in solution [64]).

#### 2.1.2. Malic Acid

Malic acid (MA, malate, 2-hydroxybutanedioic acid), a naturally occurring four-carbon dicarboxylic acid and an intermediate in the Kreb’s Cycle, is formed from fumaric acid and oxidized to oxoacetic acid. MA can also be metabolized to pyruvic acid. Like other OAs, MA forms strong bonds with HM ions via the carboxyl groups, harboring the function of donor oxygen in the metal-ligand [65], and is the predominant OA in many fruit species, including apples, stone fruits, and berries [66,67]. As a potent HM-chelator, exogenous application has been used to combat HM stress (Table 2). Indeed, in a comparative study, where *S. variegata* seedlings were exposed to contamination by Cd (50 μM as CdCl_2_), of three different OAs (MA, CA, and tartaric acid; 100 μM), the plants treated with MA had a more pronounced increase in root, stem, and total biomass, and a greater decrease in the content of MDA compared to CA- or tartaric acid-treated lines [33]. Root and leaf POD activities were also raised by addition of MA in Cd-contaminated plants [33]. Highlighting the role of MA in nurturing the recovery of physiological parameters decreased by HMs, other examples of plants treated with MA include *Miscanthus sacchariflorus* (100 μM CdCl_2_), mustard (NiSO_4_ 0.003 mM), *Alyssum corsicum* (NiSO_4_ 0.3 mM), spinach (*Spinacea oleracea*; <4.83 mM Pb), and rice (25 μM CdCl_2_) [17,26,34,35]. Other crops of commercial interest, such as maize and sunflower, have been exposed to MA (0.1 mmol kg^−1^ and <500 μM, respectively) as a means to withstand Ni (250 mg Ni kg^−1^, maize) and Cd (5 μM CdCl_2_, sunflower) [36,37]. In both cases, addition of MA increased the total biomass and the growth rate, improved the antioxidant capacity (by increasing the activity of antioxidant enzymes POD, SOD, APX, GR, etc.), and enhanced the content of photosynthetic pigments such as chlorophylls, and also anthocyanins. In addition, the expression of genes associated with HM tolerance or as indicators of HM stress (chlorophyll synthase, *CHLG*, key to the formation of chlorophyll, *OsNramp1*, *SOD*, *POD*, etc.) was higher, leading to decreased ROS levels and electron leakage (EL) in both maize and sunflower [36,37].

The effect of different OAs (CA, MA, tartaric acid) on the expression levels of genes involved in HM transport and detoxification in roots and leaves of *S. variegata* under stress by Cd (100 μM CdCl_2_) was investigated [33]. In roots, application of MA (100 μM) significantly increased the expression levels of *HMA1* and *HMA3*. In leaves, MA boosted the expression of *Nramp5* and *PCS1* to levels that were substantially higher than with the other OAs tested, whilst that of *MTP1*, *HMA3*, *HMA5*, *MT1A*, and *MT2B* fell by 37–78% compared to the Cd treatment group [33].

MA has also been shown to have the ability to increase the expression of genes encoding antioxidant enzymes. For example, in *M. sacchariflorus* subjected to stress by Cd (100 μM CdCl_2_), both the treatments by Cd alone or MA alone (100 μM) increased the expression of *Cu/Zn-SOD*, *POD1*, *glutathione peroxidase* (*GPX1*), *glutathione S-transferase* (*GST1*), *monodehydroascorbate reductase* (*MDHAR*), and *dehydroascorbate reductase* (*DHAR*). When MA and Cd were applied simultaneously, the expression of *Cu/Zn-SOD*, *POD1*, *GR1*, *GPX1*, and *GST1*, all vital for protection against oxidative stress, was induced even further, reaching up to 4.7-fold higher compared to Cd-control plants [34]. In rice *OsNramp1*, *OsIRT1* (iron regulated transporter), *OsHMA3*, and *OsNAS1* (nicotianamine synthase) had lower expression in response to stress by Cd (25 μM CdCl_2_), an effect that was reversed on the concomitant addition of MA (or CA, 50 μM), aiding in the neutralization of HM stress [26]. However, a universal increment in the expression of all antioxidant genes was not observed; for example, that of *SOD* fell on addition of MA, indicating a certain degree of specificity, probably via different pathways controlled at the transcriptional level [26].

As in the case of CA, transgenic techniques have also been explored to raise MA synthesis as a means of increasing a plant’s tolerance to HM contamination. Such is the case of transgenic tobacco overexpressing cytosolic malate dehydrogenase (*MDH*) from *Arabidopsis thaliana* and the MDH gene from *Escherichia coli* (*emdh*) [68]. The results demonstrate that genes from both origins raise MDH activity (by up to 120–130%), including greater malate concentrations in root exudates compared to non-transformed controls, when tobacco is grown under Al stress (300 μM). Such effects enhanced Al-tolerance in the transformed lines, an effect that was proportional to MA exudation [68]. Likewise, as a proof-of-concept for potential transfer to crop species, it has been demonstrated that overexpression of *MDH1* from *Stylosanthes guianensis* participates in Mn (5–800 μM MnSO_4_) detoxification by improving malate synthesis and exudation in Arabidopsis roots, thus lowering plant Mn concentrations in plants exposed to Mn stress [69]. Additionally, in the same plant model, cytosolic *MDH* (*CMDH4*) is required for Pb tolerance, enhancing the transcription of *PDR12* required for Pb efflux [70].

#### 2.1.3. Oxalic Acid

Oxalic acid (OxA) and its conjugate base oxalate can form salts with K, Cu, Mg, and Ca, the most prevalent being calcium oxalate (CaOx) [71]. OxA plays an important role in the regulation of not only Ca but also of other divalent ions and HMs (Cd, Zn, Pb, Mn), due to accumulation within the vacuoles of specialized cells called idioblasts [72]. The secretion and accumulation of oxalate by plant roots prevents the absorption of HMs, as oxalate forms complexes with different metal ions such as Pb (practically insoluble) or Cd (partially soluble), thus decreasing the bioavailability of such HMs in the soil [73]. This effect is usually attributed to a lowering in the number of ionized bioavailable forms of HMs in the rhizosphere, coupled with higher competition of protons and metal ions for the adsorption sites on root cell walls, as well as the formation of less bioavailable chelating complexes [73]. A case in mustard highlights such responses (Table 2) [38]. Specifically, anthropogenic contamination of soils with Cd and Zn led to a fall in the activity of antioxidant enzymes such as phenylalanine ammonium lyase (PAL), polyphenol oxidase (PPO), and CAT in seeds. Application of 5 mM EDTA, despite being a very effective chelator, caused negative consequences on PPO activity, plant growth, development, and shoot and root biomass. However, treatment with 5 mM OxA did not have such an effect; on the contrary, it significantly improved parameters such as the antioxidant activity of PAL, PPO, and CAT, and the dry and fresh shoot and root weights [38]. In another study, the foliar application of 2.5 mM OxA (together with indole-3 acetic acid) on *Sedum alfredii*, a perennial herb used for phytoremediation, resulted in a greater total biomass and a slight increase in the Cd or Pb content compared to the control, and improved the absorption of nutrients such as K by 38% [39]. Similarly, chickpea (*Cicer arietinum*) seeds have been bathed in an aqueous solution of 200 µM CdCl_2_, and then washed in 100 µM OxA. The stress by Cd alone considerably increased the amount of MDA and the content of H_2_O_2_ in both chickpea roots and shoots. The exogenous application of OxA significantly alleviated these molecular symptoms, returning MDA and H_2_O_2_ parameters back to non-Cd-stressed control values. A similar palliative effect was seen in the reduced/oxidized glutathione (GSH/GSSG) ratio in roots and shoots, and in GPX and GR activities, enzymes responsible for recycling antioxidant enzymes. Additionally, the balance of nicotinamides (NADP+/NAD+) and their reduced forms (NADH/NADPH) in shoots was 37% higher in chickpea seeds grown in Cd media, and the addition of OxA counteracted this adverse effect, thus restoring the balance between oxidized and reduced forms [40].

#### 2.1.4. Lipoic Acid

Lipoic acid (6,8-dithiooctanoic acid, LA) is an 8-carbon molecule, a fundamental cofactor for the activity of five enzyme complexes that are part of the central metabolism: α-ketoglutarate dehydrogenase (KGDH), pyruvate dehydrogenase (PDH), glycine decarboxylase (GDC), branched chain α-keto acid dehydrogenases (BCDH), and acetoin dehydrogenase (AoDH) [74,75]. LA and its reduced form, dihydrolipoic acid (DHLA) [41,76], have antioxidant activity and can chelate HMs; LA can chelate metals such as Mn, Cu, Zn, and Pb, while DHLA chelates Fe^2+^, Fe^3+^, Co, Ni, Cu, Zn, Pb, and Hg. Although it is well documented that LA has the ability to chelate metal ions in addition to antioxidant properties, the chelation ability of DHLA and LA, due to the presence in the molecules of both vicinal sulfur atoms and of a carboxylic group is important in biological systems since, in the presence of excess free HMs, they can participate in chain reactions, leading ultimately to O_2_ reduction by catalyzing electron transfer from one oxygen species to another in a Fenton-like reaction [76]. Although most research in plants has focused on how internal LA and DHLA levels change in response to HM stress (e.g., in durum wheat, *Triticum durum*, treated with Cu [77]), one relevant study highlights its utility as an exogenous agent (Table 2). In Turk et al. [41], the physiological and biochemical effects related to the application of LA in wheat (*Triticum aestivum*) were demonstrated. Seeds were soaked with 1.5 mM Pb (NO_3_)_2_ with or without 2 μM LA to determine whether the latter is able to mitigate the effects produced by HM contamination. The treatment with Pb significantly increased the amounts of O^2−^ and H_2_O_2_ by 42% and 58% respectively, increments which were buffered on addition of external LA (by 19% and 20%, respectively). Similar effects were seen in lipid peroxidation (determined as MDA) and SOD, although not for other enzymes with known antioxidant capacities like GPX, GR, and APX [41].

#### 2.1.5. Jasmonic Acid

Jasmonic acid (JA) and its derivatives, jasmonates, are carboxylic acids and lipid phytohormones belonging to the group of oxylipins [78]. Discovered in 1962, this phytohormone is part of processes such as adaptation to stress, reproductive growth of plants, movement, etc. [78]. Physiologically, the application of JA (and its derivatives) has been implemented to alleviate the symptoms produced by contamination by HMs, including Cr, As, Ni, Cd, Pb, and Cu (Table 2) [79]. For instance, the application of Cr (150 and 300 μM) in Choy sum (*Brassica parachinensis*) generates a drop in morphological (root length, plant height, and biomass, etc.) and biochemical (total chlorophyll and carotenoid content, etc.) parameters, and altered the activity of antioxidant enzymes (including SOD, CAT, APX, GR, GST, MDHAR, and DHAR) [46]. However, the foliar application of JA (5, 10, 20 μM) neutralized the toxic effects of Cr, particularly at the highest dose. The recovery was shown by the reversal in the falls in morphological parameters, photosynthetic pigment levels, and gas exchange, a reduction in the absorption of Cr by roots and restored mineral homeostasis (K, Ca, Mn, Fe, Zn). Exogenous JA also increased activities of the antioxidant enzymes SOD, CAT, APX, GR, GST, MDHAR, and DHAR in Cr-treated plants, mirrored by falls in MDA and H_2_O_2_ levels [46]. Similar protective effects have been observed in wheat (100 μM Cd plus methyl jasmonate (MJ, 10 μM spray) [47]), runner beans (*Phaseolus coccineus*; 50 μM Cu plus 10 μM MJ [48]), and alfalfa (100 μM Cu plus 1–10 nM JA [49]). 

In rice exposed hydroponically to Pb (150–300 μM) and MJ (0.5–1 μM), in addition to improving the physiological parameters decreased by Pb, a study also demonstrated that MJ can modulate the expression of certain genes, including *HMAs*, *PCS1* and *2*, and *ABCC1* (Cd transporter into the vacuole) [42]. The toxicity of Pb increased expression of *HMA2* (involved in the transport of Cd into shoots), *HMA3* and *HMA4* (transporters of Cd into vacuoles), *PCS1* and *2*, and *ABCC1*. When exogenous MJ was applied, the expression of *HMA2* and others decreased in roots, effectively trapping Pb underground and limiting translocation to aerial parts of the Pb-treated rice [42]. Similar protection was observed when MJ (0.5–1 μM) was added to rice supplemented with As (25–50 μM), in that MJ reduced the absorption of As into rice roots and leaves and significantly lowered the amount of MDA and H_2_O_2_ (as reflected in increased APX, SOD, and CAT activities and levels of ascorbic acid (AsA), dehydroascorbate (DHA), and GSH), thus restoring overall morphological and physiological fitness [43]. At the transcriptional level, when As was applied, the expression of As transporters (*Lsi1*, *Lsi2*, and *Lsi6*) increased significantly (by up to 6.4-fold, depending on the rice variety), effects that were reversed substantially by the exogenous supplementation of the growth medium with MJ, effectively reducing As influx into the rice plant [43].

Furthermore, when tomato seeds were soaked in solution with JA (100 nM) for 4 h to neutralize the harmful effects of Pb (0.25–0.75 mM), the results indicate that exogenous JA triggered increases in relative water content (RWC) and levels of total photosynthetic pigments, antioxidant secondary metabolites (flavonoids, anthocyanins, etc.), and OAs, including CA, MA, and other Kreb’s Cycle intermediates, compared to the controls exposed only to Pb [44]. Such alterations were accompanied by concordant changes in the levels of gene expression of *CS*, *PAL*, *CHLASE*, and *CHS*, amongst others [44].

Even in non-crop species, adding JA or its derivatives can be used to rescue phenotypes brought about by deficiencies in the biosynthesis of this phytohormone. Such is the case for the AtAOS mutants of Arabidopsis, deficient in allene oxide synthase, a key protein required for JA manufacture, whereby it was shown that in the presence of 50 mM CdCl_2_, mutants had higher expression of *AtIRT1* (Cd uptake), and *AtHMA2* and *4* (Cd translocation) reflected the exhibition of much more pronounced symptoms on exposure to Cd (chlorosis and shorter roots), and more Cd in shoots and roots [45]. Nevertheless, the molecular and morphological phenotypes could be alleviated if MJ (0.01, 0.025 mM) was simultaneously added to the growth medium of the AtAOS mutants [45]. Similar transcriptomic analyses have been performed in rice, whereby Cu (100 μM) increases the expression of genes encoding enzymes related to JA biosynthesis, including AOS [80], as does the exposure to As (5–200 µM) [81].

### 2.2. Phenolic Acids

#### 2.2.1. Salicylic Acid

Salicylic acid (SA, ortho-hydroxybenzoic acid) is a phenolic phytohormone present to a large extent in the plant kingdom and is a key regulator of processes such as thermogenesis, plant signaling, regulating plant morphology, development, flowering, and closure of the stomata, as well as participating as a mediator of the defense responses against pathogens and to abiotic stresses [82]. There are two pathways involved in SA biosynthesis—the shikimate pathway found in the cytoplasm, and the isochorismate route localized in the chloroplast [83]. One of the many abiotic stresses which SA is involved in is the contamination by HMs [82]. In general, under Cd exposure, pre-treating with SA increases the ROS-quenching capacity of contaminated plants, like maize and tomato, by stimulating the activity of antioxidant enzymes, boosting levels of non-enzymatic antioxidant compounds with HM-chelating capacities (PCs, GSH, etc.), and strengthening cell walls to limit Cd accumulation (Table 2) [55,84,85]. In more detail, the foliar application of SA (600 μM) to potato explants (*Solanum tuberosum*) treated with Cd (200 μM CdCl_2_) improved parameters reduced by stress with Cd, such as the relative water content (RWC), chlorophyll and proline levels, along with decreasing MDA and ROS [50]. In lemon balm (*Melissa officinalis*), in the presence of Ni (0–500 μM) in nutrient solution, the application of SA (1.0 mM) by manual spraying was successfully used to counteract the symptoms of Ni contamination [57]. As a genetic strategy, overexpression of bacterial salicylate hydroxylase (*NahG*), an enzyme critical in the shikimate pathway, enabled transgenic Arabidopsis to withstand the negative effects induced by watering with CdCl_2_ (0.5 mM) [86].

At the molecular level, contamination of lemon balm by Hg (50 μM) lead to a significant reduction in the expression of *CHLG* and an increase in that of *PAL*, important for the formation of phenylpropanoids and acclimatization to abiotic stress. The application of SA (50 μM) to Hg-treated lines enhanced the expression of *CHLG* and reduced that of *PAL*, suggesting that SA could play a role in the regulation of genes related to the biosynthesis of phenolic compounds [51]. In another study, where maize seeds were exposed to stress by Pb (2.5 mM), the selected genes were *ZmACS6* and *ZmSAMD* (involved in the metabolism of the ethylene precursor methionine, and in polyamine biosynthesis, respectively). Under Pb stress, these genes had raised expression levels in roots and significantly lower transcript contents in shoots. Treatment with SA (0.5 mM) reversed these trends in both aerial and underground parts of the maize plants, implicating SA in the synthesis of ethylene and in tolerance to Pb stress [87]. The importance of exogenous SA is also demonstrated in the regulation of growth, pigment content, antioxidant defense responses, and gene expression in mustard seedlings exposed to different concentrations of Pb (0.25–0.75 mM). In an analysis of *POD*, *DHAR*, *GST*, and *GR*, it turned out that contamination decreased the expression of these genes, whereas soaking seeds with 1 mM SA led to dramatic improvements (by up to 5-fold) in the transcript levels of all the genes involved in oxidative stress responses [53].

Not all genes regulated by SA are related to oxidative stress. In an interesting recent development, researchers demonstrated that cell wall components also play a key role. Specifically, pretreatment of tomato leaves by spraying with SA (100 μM) had significant priming effects prior to exposure to Cd (100 μM for 3 days, then 10 μM in hydroponic medium). SA pretreatment followed by Cd addition reduced the activity of pectin methylesterase (PME), an enzyme that catalyzes the demethylation of pectin, thereby exposing a large area of free carboxyl groups to bind metal ions by approximately a third in both roots and leaves, changes reflected in the transcript levels of *PME1* and *2* [55]. In both organs, SA-pretreated Cd plants also had greater cell wall peroxidase and laccase capacity, and expression of *TAP2* (peroxidase) and *LAC* (laccase) compared to tomato exposed solely to Cd. Possibly reflecting greater changes in cell wall structure, genes required for cellulose synthesis were also upregulated in tomato pretreated with SA, revealing concerted responses to reduce potential loading of Cd into the plant symplast [55].

#### 2.2.2. Gallic Acid

Gallic acid (GA; 3,4,5-triphydroxyl-benzoic acid) is a widely distributed phenolic acid in many different families of plants and is present in different parts of these, such as roots, leaves, seeds, stem, woody tissues, and especially fruits like raspberries, blueberries, and strawberries [88]. The protective effect of GA (and phenolic acids in general) is correlated with their antioxidant and free radical scavenging power, their potential ability to interact with transduction signal pathway components, and their HM chelating activity [89]. GA is a secondary metabolite of plants and arises mainly from the shikimate route, from 3-dehydroshikimic acid, 3-DHS [90]. Subsequently, dehydroquinate dehydratase (DQD)/shikimate dehydrogenase (SDH) oxidizes 3-DHS to shikimate acid (DQD activity) or reduces the same substrate to GA via the SDH domain [91]. Indeed, overexpression of SDH from walnut (*Juglans regia*), a species that produces very high concentrations of GA, in tobacco (*N. tabacum*), lead to transformants accumulating >500% more GA than the controls [91]. In addition, GA itself can act as a natural fertilizer (without contamination by HM), especially because it is an excellent neutralizer of ROS (both directly, and by enhancing antioxidant activity), with positive impacts on morphological parameters like germination rate, plant biomass, and yield. For instance, adding GA (60 µg mL^−1^) to rice increased the expression of genes related to antioxidant activity, such as *PAL* and *CHS*, more than 4-fold [92].

Evidence pointing to a role of GA in HM responses shows that diverse metals in the growth substrate can trigger short-term increases in this phenolic acid (Table 2). For instance, exposure of bean sprout roots (*Phaseolus vulgaris*) to Pb and Cd (1–10 μM) had an initial positive impact on SDH activity in roots and leaves for three days before inhibitive effects (either of the HM itself, or from accumulated ROS) caused SDH activity to fall back [93].

Regarding the contamination by HMs, GA can alleviate the physiological symptoms produced by Cd stress in sunflower seeds. In these recent experiments, irrigation with 20 μM CdCl_2_ caused increases in endogenous H_2_O_2_ levels, lipid peroxidation, EL, and SOD activity, whilst simultaneously total thiols and chlorophyll content fell, as did neutral lipids, phospholipids, and galactolipids, among other biochemical and molecular parameters [59]. However, pretreatment of the seeds with GA (75 μM) improved and/or reverted all these symptoms. Higher chlorophyll, total thiol, neutral lipids, phospholipids, and galactolipids contents were recorded, whilst H_2_O_2_, MDA, EL, and SOD levels all dropped [93].

Similar results were obtained in maize shoots grown in Cu (1 mM). The application of GA (1.5 mM) reverted the symptoms caused by this HM, such as a reduction of RWC, and SOD, CAT, and APX activities, whilst proline and H_2_O_2_ levels were higher [60]. Likewise, exogenous application of 25 µM GA to wheat (*T. aestivum*) cultivated in Cd (100–200 µM) and redressed the misbalance of the ascorbate-GSH cycle, showing the effectiveness of this phenolic acid in maintaining antioxidant capacity [61].

#### 2.2.3. Caffeic Acid

Caffeic acid (CfA; 3,4-dihydroxycinnamic acid) is a cinnamic acid and a phenylpropanoid present in all plants known to be involved in lignin synthesis, but also plays roles in cell expansion (and therefore growth), turgor, and phototropism. CfA and its derivatives are known for their antioxidant power and their role in tolerance to biotic and abiotic stresses, the latter including temperature, osmotic, and HM challenge [94]. CfA is an intermediate of the phenylpropanoid pathway synthesized from p-coumarate from phenylalanine or L-tyrosine, by PAL/cinnamate 4-hydroxylase, or tyrosine ammonia-lyase, respectively. Subsequently, p-coumarate is hydroxylated to CfA [95]. It has been proposed that the protective capabilities of CfA in HM stress are due to the generation of changes in the cell wall increasing its durability (probably via the production and accumulation of more lignin) and/or the direct neutralization of ROS, thus preventing damage to cellular macromolecules [96,97].

However, several studies have focused on the exogenous application of CfA on plant responses to HM stress due to their presence in root exudates (Table 2) [98]. Regarding a primary response of crops subjected to HM stress, it is known that HMs in the growth medium can drive production of CfA and its ester, rosmarinic acid (caffeic acid ester), in a manner that is directly proportional to the intensity of the stress, as shown using Cd (0.2–1.8 mmol L^−1^), Pb, and Al (0.04–0.16 mmol L^−1^) [99]. It is also important to note that CfA is a critical precursor of other chemical compounds that have fundamental roles in tolerance to stress by HM, such as ferulic acid and melatonin [100,101]. Indeed, a key enzyme for the conversion to both derivatives is caffeic acid O-methyltransferase (*COMT*). Gene expression of *COMT* is significantly increased in alfalfa seeds grown under stress by Cu and Cd (250 μM) [62], as is that of several members (*BnCOMT*-*1*, -*5* and -*8*) of the COMT gene family in canola sprouts (*Brassica napus*) cultivated in Al and Cd (25 μM) [102].

## 3. Conclusions and Future Perspectives

Considering that the World Health Organization (WHO) establishes that several HMs constitute a major public health concern (As, Cd, Hg, and Pb) [103], the problems affecting plant, human, and animal health have tended to increase due to greater soil and bedrock disturbance, pesticide use, chemical disposal, etc., especially in the last 200 years. Although biotechnological advances, including transgenic strategies, have undoubtedly led to great strides in improving the resilience of crop plants to HM stress (e.g., [104,105,106,107,108,109]), the exogenous application of OAs serves as a viable solution in both economic and eco-friendly terms as a means of using a nature-based solution to contamination caused both naturally and anthropogenically. As primary or secondary metabolites, the responses of plants that are under stress by HMs are alleviated by OAs (Figure 1), as visualized by (I) an increase in morphological parameters such as growth, biomass, leaf area, and root length, leading ultimately to greater crop productivity, (II) an increment of the total amounts of chlorophyll and other photosynthetic pigments, (III) a boost in the activity of antioxidant enzymes SOD, POD, APX, GR, CAT, DHAR, MDHAR, PAL, PPO, etc., (IV) an enhancement in the activity of non-enzymatic antioxidant components including AsA, tocopherol, GSH, etc., (V) a fall in ROS, lipid damage, and EL, and (VI) HM chelation. In parallel, OAs can affect the expression of certain groups of genes, such as those (I) directly-required for OA biosynthesis (e.g., *CS*, *malate synthase*, etc.), (II) needed for the synthesis of protein-based HM-chelators (PCs, MTs, etc.), (III) involved in specific HM transport across the plasma membrane or tonoplast (IRT-like proteins, NRAMPs, HMAs, MTPs, etc.), and (IV) encoding antioxidant enzymes [104].

Taken together, in plants, fungi, and microorganisms, more than 200,000 primary and secondary metabolites are synthesized [110], of which a substantial proportion correspond to carboxylic and phenolic acids. Studying the individual influences of each OA thus poses a clear challenge to scientists. Such a challenge is compounded by the fact that some effects on alleviating HM contamination are shared by several crop species studied, whereas the intensity of responses can even depend on the genotype used in the research (e.g., rice in [43]). However, without doubt, many of the differences observed in experimental outcomes using the same OA can be attributed to experimental design. As shown in Table 2, the intensity of stress (HM concentration), the site and method of OA application (foliar or growth medium, spray, or irrigation), the concentration of OA used, and the timing of its addition (seed imbibition, treatments prior to and/or concurrently with HM stress) play a role in determining the findings, and thus the conclusions drawn for the potential translatability of the research to field settings.

A further development is the application of combinations of plant OAs. For example, the effects of the presence of both CA and MA (50 μM in nutrient medium) have been evaluated in the rice treated with Cd (25 μM CdCl_2_, in nutritive medium). In the presence of OAs (without Cd), the expression of *OsNramp1*, *OsIRT1*, *OsHMA3*, and *OsNAS1* decline, including those for vacuolar Cd transport. However, the addition of both OAs neutralized the adverse effects of Cd on the expression of these host genes, improving sequestration of Cd in the root vacuole, boosting antioxidant levels and overall photosynthetic efficacy, leading to a general increase in biomass [26].

Finally, interdisciplinary research is bringing together researchers from different areas, thus providing additional strategies to alleviate common agricultural problems. For example, the OA phytic acid (myo-inositol) plays a fundamental role as a reservoir of phosphate since it has six phosphate groups, yet it also participates as a divalent cation chelator given its strong negative charge [111]. However, as phytic acid can play a role as a chelator of HMs in in vitro tests [112], a novel application includes the design of electrospun polyurethane/phytic acid nanofibrous membranes for adsorbing Pb [113]. These results demonstrate that this plant-derived OA can indeed remove Pb from an aqueous solution. Further collaborative research and the possible sequestering of this and other HMs in field situations are thus eagerly-awaited developments.

## Figures and Tables

**Figure 1 ijms-23-05438-f001:**
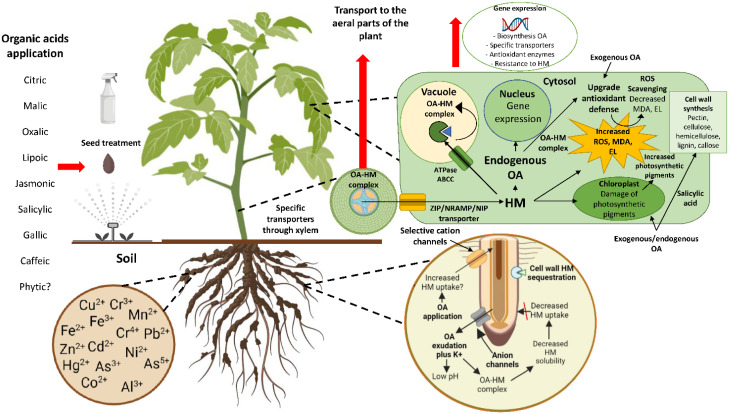
Potential molecular and physiological mechanisms triggered by the exogenous application of organic acids that increase the tolerance of plants to heavy metal stress.

**Table 1 ijms-23-05438-t001:** The chemical structures of organic acids presented in this review.

Organic Acid	Structure	IUPAC Name
Citric acid	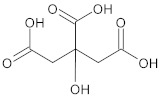	2-hydroxypropane-1,2,3-tricarboxylic acid
Malic acid	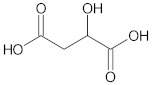	2-hydroxybutanedioic acid
Oxalic acid	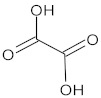	1,2-ethanedioic acid
Lipoic acid	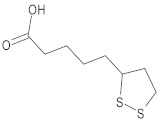	5-[(3*R*)-dithiolan-3-yl]pentanoic acid
Jasmonic acid	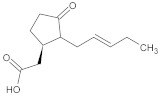	2-[(1*R*,2*R*)-3-oxo-2-[(*Z*)-pent-2-enyl]cyclopentyl]acetic acid
Salicylic acid	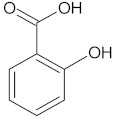	2-hydroxybenzoic acid
Gallic acid	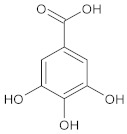	3,4,5-trihydroxybenzoic acid
Caffeic acid	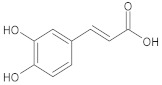	(*E*)-3-(3,4-dihydroxyphenyl)prop-2-enoic acid

**Table 2 ijms-23-05438-t002:** The molecular and physiological effects of exogenous organic acid applications to crop species subjected to heavy metal stress.

Organic Acid	Species	HM Stress	OA Application	Gene Expression Responses *	Physiological and Morphological Outcomes	Reference
Citric acid	*Brassica juncea*	Cd as CdCl_2_ (0.6 mM)	0.6 mM in soil	Up-regulation: *PSY*, CHS. Down-regulation: *CHLASE*.	Increased: Growth, biomass, total chlorophyll, carotenoids, anthocyanins, flavonoids, gaseous exchange parameters, activities of antioxidant enzymes SOD, POD, CAT, GPOX. Decreased: MDA.	[23]
	*Oryza sativa*	Cd as CdCl_2_ (25 μM)	50 μM in nutritive medium	Up-regulation: *OsNramp1*, *OsIRT1*, *OsHMA3*, *OsNAS1*. Down-regulation: *OsSOD*, *OsCAT*.	Increased: Biomass, photosynthetic pigments, activities of antioxidant enzymes. Decreased: Cd content in leaves	[26]
	*Oryza sativa*	Cd as CdCl_2_ (0.1, 0.6, 0.9, 1.2, 2.4 mg kg^−1^)	5 mM by spraying	Up-regulation of *OsNramp1*, *2*, *3*, *5*.	Increased: Content of Glu, Phe, His, Ser and Thr, Arg; Mn mobilization, Mn/Cd ratio. Decreased: Cd mobilization.	[32]
	*Salix variegata*	Cd as CdCl_2_ (50 μM)	100 μM in aqueous solution	Up-regulation: *HMA1*, *PCS1*, *HMA3*, *Nramp5*, *MTP1*, *MTP4*, *HMA3*, *MT1A*, *MT2B*.	Increased: Growth, biomass, activities of antioxidant enzymes SOD, POD, CAT, APX; Non-Protein Sulfhydryl compounds (NPT), GSH, and non-GSH NPT. Decreased: MDA.	[33]
	*Solanum lycopersicum*	Pb as Pb(NO_3_)_2_ or As as Na_2_HAsO_4_ (10 μM)	250 μM in nutritive solution	-	Increased: Growth rate, photosynthetic pigments, activities of antioxidant enzymes CAT, APX, GR. Decreased: MDA, DNA damage.	[18]
	*Brassica napus*	Cu as CuSO_4_ (50, 100 μM)	2.5, 5 mM in nutritive medium	-	Increased: Biomass, photosynthetic pigment, activities of antioxidant enzymes CAT, POX, SOD. Decreased: MDA, EL, H_2_O_2_.	[14]
	*Brassica juncea*	Cr (100, 500 μM) in solution	2.5, 5 mM in nutritive medium	-	Increased: Biomass, photosynthetic pigments, activities of antioxidant enzymes SOD, POD, CAT, APX. Decreased: ROS, MDA.	[16]
	*Solanum nigram*	Cd (50 mg Cd^2+^) in dry soil	20 mM in dry soil	-	Increased: Growth, biomass, plant weight, activities of antioxidant enzymes SOD, POD. Decreased: MDA.	[24]
	*Zea mays*	Cd as CdCl_2_ (300 mg kg ^−1^)	0.25, 0.5, 1, 2 g kg^−1^ by irrigation	-	Increased: Biomass, shoot and root length. Decreased: Cd uptake.	[25]
	*Helianthus annuus*	Cr (5, 10, 20 mg kg^−1^) in dry soil	2.5, 5 mM	-	Increased: Growth, biomass, photosynthetic pigments, activities of antioxidant enzymes. Decreased: ROS, MDA.	[19]
	*Calendula officinalis*	Cd-spiked soils (50, 100 mg kg^−1^)	0.05, 0.1 mM in soil	-	Increased: Root and shoot dry weight, photosynthetic pigments, activities of antioxidant enzymes SOD, CAT, APX. Decreased: MDA, H_2_O_2_.	[29]
	*Salix viminalis*	Cd as Cd(NO_3_)_2_ (3, 6 mg kg^−1^) in spray	20 mM in aqueous solution	-	Increased: Biomass, Cd plant uptake, photosynthetic pigments, leaf gas exchange, photosynthetic rate. Decreased: Pro content.	[30]
	*Larix olgensis*	Pb as Pb (NO_3_)_2_ (100 mg kg^−1^)	0.2, 1, 5, 10 mM by irrigation and leaf spray	-	Increased: Survival rate, biomass, photosynthetic pigments, activities of antioxidant enzymes SOD, POX, Pro content. Decreased: Pb content, MDA.	[31]
	*Typha latifolia*	Pb as Pb(NO_3_)_2_ and Hg as HgSO_4_ (0, 1, 2.5, 5 mM) in nutritive medium	5 mM in nutritive medium	-	Increased: root, stem and leaf biomass, leaf number and areas, plant height, root length, photosynthetic pigments, activities of antioxidant enzymes SOD, POX, CAT, APX. Decreased: MDA, EL, ROS.	[19]
Malic acid	*Salix variegata*	Cd as CdCl_2_ (50 μM) in aqueous solution	100 μM in aqueous solution	Up-regulation: *HMA1*, *PCS1*, *HMA3*, *Nramp5*, *MTP4*. Down-regulation: *MTP1*, *HMA3*, *MT1A*.	Increased: Growth, biomass, activities of antioxidant enzymes SOD, POD, CAT, APX; non-protein sulfhydryl compounds (NPT), GSH, non-GSH NPT. Decreased: MDA.	[33]
	*Miscanthus Sacchariflorus*	Cd as CdCl_2_ (100 μM) in nutritive solution	100 μM in nutritive solution	Up-regulation: *Cu/Zn-SOD*, *POD1*, *GPX1*, *GST1*, *MDHAR*, *DHAR*. Down-regulation: *CAT1*.	Increased: Growth, root and shoot length, photosynthetic pigments, total antioxidant capacity, activities of antioxidant enzymes SOD, CAT, POD, APX, GR, GPX, and GST; concentration GSH and GSSG. Decreased: MDA, ROS	[34]
	*Oryza sativa*	Cd as CdCl_2_ (25 μM)	50 μM in nutritive solution	Up-regulation: *OsCDT1*, *OsNramp1*, *OsIRT1*, *HMA3*. Down-regulation: *OsNAS1*, *OsSOD*.	Increased: Biomass, photosynthetic pigments, activities of antioxidant enzymes. Decreased: Cd content in leaves.	[26]
	*Brassica juncea*	Ni as NiSO_4_ (0.003 mM) in nutritive solution	0.5, 1, 5 mM in nutritive solution	-	Increased: Ni leaf concentration. Decreased: Leaf biomass, Ni root uptake.	[17]
	*Alyssum corsicum*	Ni as NiSO_4_ (0.3 mM) in nutritive solution	0.5, 1, 5 mM in nutritive solution	-	Increased: Shoot and root biomass. Decreased: Ni shoot concentration.	[17]
	*Spinacea oleracea*	Pb (2.42, 4.83 mM) in nutritive solution	2.4 mM in nutritive solution	-	Increased: Biomass, shoot length, photosynthetic pigments, activities of antioxidant enzymes SOD, GPOX, CAT, APX, AsA contents, total phenolics. Decreased: MDA, ROS, flavonoid content.	[35]
	*Zea mays*	Soil polluted with 250 mg Ni kg^−1^	0.1 mM in nutritive solution	-	Increased: Shoot dry weight, Ni uptake efficiency (without soil P). Decreased: Ni uptake efficiency (with soil P).	[36]
	*Helianthus annuus*	Cd as CdCl_2_ (5 μM) in nutritive solution	250, 500 μM in nutritive solution	-	Increased: Growth, biomass, shoot and root length, photosynthetic pigments, OA content, activities of root dehydrogenases. Decreased: ROS, H_2_O_2_.	[37]
Oxalic acid	*Brassica juncea*	Cd and Zn resulted from smelting waste emissions	Drip irrigation system (5 mM)	-	Increased: Biomass, root and shoot dry weight, Zn and Cd mobilization, activities of antioxidant enzymes of PAL, PPO, and CAT.	[38]
	*Sedum alfredii*	Cd (10.71 mg kg^−1^) and Pb (438.4 mg kg^−1^) in contaminated soil	2.5 mM by leaf spray	-	Increased: Biomass, plant growth, Cd and Pb mobilization, photosynthetic pigments, K content. Decreased: MDA.	[39]
	*Cicer arietinum*	Cd as CdCl_2_ (200 µM) by seed imbibition	100 μM in aqueous solution	-	Increased: root and shoot growth, activities of antioxidant enzymes GPX, GR, glutathione redox state, NADP+/NAD+ ratio, NADH+ NADPH ratio. Decreased: MDA, ROS, carbonyl group contents.	[40]
Lipoic acid	*Triticum aestivum*	Pb as Pb(NO_3_)_2_ (1.5 mM) by seed imbibition	2 μM by seed imbibition	-	Increased: Enzymatic activity amylase, SOD, GSH, GSH/GSSH ratio. Decreased: O^2−^ y H_2_O_2_.	[41]
Jasmonic acid	*Oryza sativa*	Pb as Pb(NO_3_)_2_ (150, 300 μM) in hydroponic solution	0.5, 1 μM in hydroponic solution	Up-regulation: *HMA3*, *HMA4*, *PCS1*, *PCS2*, *ABCC1*. Down-regulation: *HMA2*.	Increased: Growth, photosynthetic pigments, Pro. Decreased: MDA, ROS.	[42]
	*Oryza sativa*	As (0, 25, 50 µM) in hydroponic solution	0.5, 1 µM MJ in hydroponic solution	Up-regulation: *IRO6*, *FRDL1*, *YSL2*. Down-regulation: *Lsi1*, Lsi2, *Lsi6*, *Nramp1*, *Nramp5*.	Increased: Height, biomass, photosynthetic pigments, endogenous JA content, activities of antioxidant enzymes CAT, SOD, APX, POD. Decreased: MDA, ROS, As concentration in roots and leaves.	[43]
	*Solanum lycopersicum*	Pb (0, 0.25, 0.50, 0.75 mM) on filter paper	100 nM by seed imbibition	Up-regulation: *succinyl CoA ligase*, *succinate dehydrogenase*, *fumarate hydratase*, *CHS*, *PAL*. Down-regulation: *CHLASE*, *CS*, *malate synthase*.	Increased: RWC, photosynthetic pigments, antioxidant molecules. Decreased: Pb concentration.	[44]
	*Arabidopsis thaliana*	Cd as CdCl_2_ (50 μM) in nutrient solution	0.01, 0.025 μM MJ in nutrient solution	Down-regulation: *AtIRT1*, *AtHMA2*, *AtHMA4*.	Increased: Cd content in root cell wall. Decreased: chlorosis, Cd content in shoot and root cell sap.	[45]
	*Brassica parachinensis*	Cr as K_2_Cr_2_O_7_ (150, 300 μM) in solution	5, 10, 20 µM by leaf spray	-	Increased: Growth, biomass, plant height, leaf area and number, photosynthetic pigments, activities of antioxidant enzymes SOD, APX, CAT, GPX, GST, GR, MDHAR, DHAR, AsA, and GSH contents. Decreased: MDA, ROS, Cr uptake.	[46]
	*Triticum aestivum*	Cd as CdCl_2_ (100 μM) in solution	10 μM MJ by leaf spray	-	Increased: Growth, biomass, RWC, photosynthetic pigments, activities of antioxidant enzymes CAT, SOD. Decreased: MDA, ROS, chlorosis.	[47]
	*Phaseolus coccineus*	Cu as CuSO_4_ (50 μM) in hydroponic solution	10 mM MJ preincubation in hydroponic solution	-	Increased: Activities of antioxidant enzymes CAT, APX, POX. Decreased: MDA, ROS.	[48]
	*Medicago sativa*	Cu as CuSO_4_ (100 µM) in nutritive medium	1, 5, 10 nM MJ in nutritive medium	-	Increased: Biomass, photosynthetic pigments, activities of antioxidant enzymes CAT, SOD, POD, APX GR. Decreased: MDA, ROS, Cu concentration in roots and leaves.	[49]
Salicylic acid	*Solanum tuberosum*	Cd as CdCl_2_ (200 μM)	600 μM by leaf spray	Up-regulation: *StSABP2*, *StSOD*, *StAPX*.	Increased: RWC, photosynthetic pigments, Pro and SA content. Decreased: MDA, H_2_O_2_, O^2^.	[50]
	*Melissa officinalis*	Hg as HgCl_2_ (50 μM) in nutritive solution	50 μM in nutritive solution	Up-regulation: *CHLG*, *PAL*	Increased: Growth, biomass, RWC, photosynthetic pigments, total phenolics, antioxidant activities, Pro content. Decreased: MDA, ROS.	[51]
	*Hordeum vulgare*	Cd (25 μM) in hydroponic culture	500 μM priming of dry caryopses	Up-regulation: *GS*	Increased: Growth, fresh and dry weight of roots and shoots, antioxidant activities CAT, APX, GPX Decreased: MDA.	[52]
	*Zea mays*	Pb as Pb(NO_3_)_2_ (2.5 mM)	0.5 mM pretreated seed	Up-regulation: *ZmACS6*, *ZmSAMD*.	Increased: Glycine betaine and nitric oxide contents. Decreased: Met, Arg, Pro contents.	[51]
	*Brassica juncea*	Pb (0.25, 0.50, 0.75 mM) in solution	1 mM by seed imbibition	Up-regulation: *PSY*, *CAT*, *POD*, *DHAR*, *GST*, *GR*. Down regulation: *CHLASE*.	Increased: Growth, root and shoot length, photosynthetic pigments, activities of antioxidant enzymes POD, APOX, GR, DHAR, MDHAR, GST, and GR, activities non-enzymatic antioxidants glutathione, ascorbic acid tocopherol. Decreased: ROS.	[53]
	*Artemisia annua*	As as Na_2_HAsO_4_ (100, 150 μM)	100 μM in nutritive solution	Up-regulation: *ADS*, *CYP71AV1*, *DBR2*, *ALDH1*.	Increased: Growth, biomass, photosynthetic pigments, activities of antioxidant enzymes SOD, CAT, APX, GR artemisinin and dihydroartemisinin. Decreased: ROS.	[54]
	*Solanum lycopersicum*	Cd (10 μM) in pretreatment and hydroponic culture	25, 50, 100, 200 μM in pretreatment and leaf spray	Up-regulation: *TAP2*, *LAC*, *CesA1*, *CesA6*. Down-regulation: *PME1*, *PME2*.	Increased: Pectin, cellulose, hemicellulose, lignin and callose synthesis in root and leaf cell wall. Decreased: Cd accumulation in cell wall, cytoplasm, organelles.	[55]
	*Lemna minor*	Cd as (10 μM Cd^2+^) in nutritive medium	50 μM in nutritive medium	-	Increased: Fe, Mg, Ca, Mo, photosynthetic pigments, activities of antioxidant enzymes SOD, GPX, CAT, APX, GR, endogenous SA, and PAL activity. Decreased: Chlorosis, MDA, ROS, ascorbate, Pro.	[56]
	*Melissa officinalis*	Ni as NiCl_2_ (500 μM) in nutritive solution	1 mM by leaf spray	-	Increased: Growth, shoot and root fresh and dry weights, photosynthetic pigments, root Pro content. Decreased: leaf Pro content, MDA, H_2_O_2_, EL.	[57]
	*Sorghum bicolor*	Cr as potassium dichromate (1.0, 2.0, 4.0 mg kg^−1^ soil)	0.5 nM pretreatment and leaf spray	-	Increased: Growth, number of leaves, activities of antioxidant enzymes POX, APX. Decreased: MDA, ROS.	[58]
Gallic acid	*Helianthus annuus*	Cd as CdCl_2_ (5, 10, 15, 20, 50, 100 μM) in nutritive solution	75 μM by seed imbibition	-	Increased: Growth, biomass, photosynthetic pigments, activities of antioxidant enzymes CAT, APX, SOD, GR; leaf lipid and fatty acid composition. Decreased: MDA, ROS, EL, Cd concentration in roots and leaves.	[59]
	*Zea mays*	Cu as CuSO_4_ (1 mM) by seed imbibition	1.5 mM by seed imbibition	-	Increased: photosynthetic pigments, Cu content, Pro, activities of antioxidant enzymes GPX, CAT, SOD, APX. Decreased: MDA, ROS.	[60]
	*Triticum aestivum*	Cd (100, 200, 300 μM) in nutritive solution	25, 75 μM, 1 mM in nutritive solution	-	Increased: Growth, Pro, activities of antioxidant enzymes SOD CAT, POX APX, Gr, NOX, MDHAR, DHAR¸ activities non-enzymatic antioxidants GSH, GSSG, AsA. Decreased: MDA.	[61]
Caffeic acid	*Medicago sativa*	Cu as CuSO_4_ (250 µM) and Cd as CdCl_2_ (250 µM) in solution	-	Up-regulation: *COMT* in Cd stress	-	[62]

* (-) means no changes in gene expression were measured.

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
