# Peer review of "Increasing Heavy Metal Tolerance by the Exogenous Application of Organic Acids"

_ijms, 2022, doi:10.3390/ijms23105438_

Round 1

Reviewer 1 Report

Comment on IJMS MDPI

Increasing heavy metal tolerance by the exogenous application of organic acids

This is a well written review paper.

However, when simply searching in Google, we can easily find the papers with almost similar or related titles:

  1. Osmolovskaya, N., Dung, V. V., and Kuchaeva, L. 2018. The role of organic acids in heavy metal tolerance in plants. Bio. Comm. 63(1): 9–16. https://doi. org/10.21638/spbu03.2018.103.
  2. Zhang, S., Chen, H., He, D., He, X., Yan, Y., Wu, K., & Wei, H. (2020). Effects of Exogenous Organic Acids on Cd Tolerance Mechanism of Salix variegata Under Cd Stress. Frontiers in plant science11, 594352. https://doi.org/10.3389/fpls.2020.594352.
  3. Guo Yu1 , Jianchu Ma1 , Pingping Jiang1*, Jieyue Li1 , Junyu Gao1 , Shixuan Qiao1 , Zhiyong Zhao1The Mechanism of Plant Resistance to Heavy Metal. IOP Conf. Series: Earth and Environmental Science 310 (2019) 052004 IOP Publishing doi:10.1088/1755-1315/310/5/052004.

You said ‘the aim of this review is to present a recapitulation of the latest advances concerning the exogenous application of selected naturally-occurring organic acids (OAs) to crops, the chemical structures of which are shown in Table 1. In each case, their effects at morphological, physiological and molecular levels, and how exogenous supplementation has been used in crop species to mitigate the effects of excess HM levels, are discussed.’

[Comment: What are the new insights of thus review ?

What is the synthesis of all the review of this paper?

What is the main novelty of this review?

Reviewer 2 Report

I recommend that authors should focus on the detailed mechanisms involved regarding the heavy metals tolerance by exogenous application of organic acids. Moreover, authors should cite more studies and must focus on the recent ones. 

Reviewer 3 Report

Title of manuscript to accept

From what I’ve read, the manuscript seems to be very good

But ....

  1. 53 - Please add paper about photosynthesis ..

The soil composition is / was the primary factor affecting chlorophyll fluorescence (CF) parameters......

.....redirection of photosynthates towards protective mechanisms against toxic effects of metals.

for instance

Bączek-Kwinta R., Antonkiewicz J., Łopata-Stasiak A., Kępka W. 2019. Smoke compounds aggravate stress inflicted on Brassica seedlings by unfavourable soil conditions. Photosynthetica, 57, 1, 1-8. DOI: 10.32615/ps.2019.026

Please add figure about effect of heavy metals on physiological parameters .....
